# *Cremastra appendiculata* Polysaccharides Alleviate Neurodegenerative Diseases in *Caenorhabditis elegans*: Targeting Amyloid-β Toxicity, Tau Toxicity and Oxidative Stress

**DOI:** 10.3390/ijms26083900

**Published:** 2025-04-20

**Authors:** Huaying Xu, Qian Wang, Yihan Zhou, Haiyu Chen, Jin Tao, Jing Huang, Yuzhi Miao, Jiayuan Zhao, Yanan Wang

**Affiliations:** 1Key Laboratory of Land Resources Evaluation and Monitoring in Southwest, Ministry of Education, Sichuan Normal University, Chengdu 610101, China; huayingx@stu.sicnu.edu.cn; 2School of Life Sciences, Sichuan Normal University, Chengdu 610101, China; qyumengq@163.com (Q.W.); 15198074767@163.com (Y.Z.); chenhaiyu@stu.sicnu.edu.cn (H.C.); 15596896957@163.com (J.T.); 17778670810@163.com (J.H.); skymyz88@163.com (Y.M.)

**Keywords:** *Cremastra appendiculata* polysaccharides, Alzheimer’s disease, antioxidants, *Caenorhabditis elegans*, oxidative stress

## Abstract

Alzheimer’s disease (AD) is characterized by oxidative stress, amyloid-beta (Aβ) deposition, and tau hyperphosphorylation. While polysaccharides have demonstrated anti-AD effects, the properties of *Cremastra appendiculata* polysaccharides (CAPs) remain underexplored. This study evaluates the physicochemical properties, antioxidant activity, anti-AD effects, and underlying mechanisms of CAP in vitro and in *Caenorhabditis elegans* (*C. elegans*) AD models. CAP, containing 22.37% uronic acid, is stable below 270 °C and adopts a triple helix structure. Scanning electron microscopy (SEM) reveals an irregular layered architecture. In vitro, CAP exhibits significant antioxidant activity, protecting PC12 cells from Aβ-induced cytotoxicity. In *C. elegans*, CAP extends the lifespan in a concentration-dependent manner without affecting growth, alleviating tau-induced locomotor defects, reducing Aβ-induced paralysis and serotonin hypersensitivity, and decreasing Aβ deposition by 79.96% at 2.0 mg/mL. CAP enhances antioxidant capacity and heat resistance by reducing reactive oxygen species (ROS) levels and increasing glutathione S-transferase 4 (GST-4) and glutathione peroxidase (GSH-Px) activities. Additionally, CAP upregulates key genes in the insulin/insulin-like growth factor signaling pathway, including *daf-16* and *skn-1*, along with their downstream targets (*sod-3*, *ctl-1*, *gst-4*, *hsp-70*). These findings suggest that CAP has potent antioxidant and anti-AD effects, alleviating Aβ- and tau-induced toxicity, and may serve as a promising therapeutic agent for Alzheimer’s disease.

## 1. Introduction

Neurodegenerative diseases are chronic neurological disorders characterized by progressive degeneration and the functional loss of neuronal cells [1]. Alzheimer’s disease (AD), the most common progressive neurodegenerative disorder, is pathologically characterized by the accumulation of amyloid-beta (Aβ) plaque and neurofibrillary tangles (NFTs) in the brain [2]. Clinically, AD manifests as significant memory loss, motor dysfunction, and cognitive decline [3]. The central nervous system (CNS), which consumes approximately 20% of the body’s oxygen, is particularly susceptible to oxidative stress due to the relatively low antioxidant levels of neurons [4,5]. Therefore, oxidative stress levels are considered to be an important causative factor in neurodegenerative diseases, including AD [6]. Moreover, oxidative stress is also associated with multiple pathogenic processes in AD, promoting Aβ production and tau protein hyperphosphorylation, thereby accelerating amyloid plaque and the formation of NFTs [7,8,9]. Current treatments for AD, such as cholinesterase inhibitors, primarily slow disease progression but are associated with significant side effects and only alleviate symptoms without providing a cure [10]. Therefore, developing novel AD treatments with higher efficacy and fewer adverse effects is crucial.

*Caenorhabditis elegans* (*C. elegans*) is a species of soil-dwelling nematodes and is widely used as a model organism for drug screening due to its short life cycle, high reproductive rate, ease of genetic manipulation, and conserved genetic pathways with humans [11,12]. With a simple nervous system comprising 302 neurons that share many functional characteristics with mammalian neurons, *C. elegans* is frequently employed to study neurodegenerative diseases such as AD and Parkinson’s disease [13]. Numerous studies have utilized *C. elegans* AD models to identify potential anti-AD compounds [14,15,16].

Natural polysaccharides have attracted attention for their antioxidant, anti-inflammatory, and safety profiles. Studies have shown that polysaccharides can target the processing, aggregation, and clearance of misfolded Aβ and tau proteins while regulating oxidative stress and inflammation [17]. Several polysaccharides exhibit anti-AD activity. Polysaccharides from *Angelica sinensis* improved memory impairment in mice with AD by balancing free radical metabolism, reducing inflammation, and inhibiting neuronal apoptosis [18]. Similarly, *Cibotium barometz* polysaccharide enhanced antioxidant enzyme activities and reduced Aβ-induced paralysis and oxidative damage in *C. elegans* via the JNK-MAPK signaling pathway [19]. *Cremastra appendiculata*, a traditional Chinese herb derived from the dried pseudobulbs of *Cremastra appendiculata* (D. Don) Makino, *Pleione bulbocodioides* (Franch) Rolfe, and *Pleione yunnanensis* Rolfe, is rich in polysaccharides [20]. It has also been shown that *Cremastra appendiculata* polysaccharides (CAPs) exhibit antioxidant, anti-aging, anti-tumor, and immunomodulatory properties [21,22,23]. Li et al. reported that the 95% ethanol extract of *Cremastra appendiculata* inhibits the aggregation of Aβ_1–42_ peptide, a hallmark pathological protein of AD, with an inhibition rate of 74.09% at 100 μg/mL, suggesting the potential therapeutic effects of *Cremastra appendiculata* on AD [24]. However, the potential effects of CAP on AD have not been extensively investigated.

Building on our previous findings that CAP enhances antioxidant capacity and extends the lifespan in wild-type *C. elegans* [21], in this study, we further determined the physicochemical properties of CAP, such as its chemical composition, microscopic morphology, and thermal properties, etc., and utilized Aβ_1–42_ transgenic *C. elegans* (CL4176) and tau transgenic *C. elegans* (BR5270) as models to investigate the effects of CAP on physiological phenotypes, Aβ_1–42_ and tau-induced AD phenotypes, antioxidant activity, and gene expression. The aim of the present study is to elucidate the multi-target effects and mechanisms of CAP in alleviating neurodegenerative diseases, providing a theoretical basis for the development of CAP as a neuroprotective drug.

## 2. Results

### 2.1. Physicochemical Characterization of Cremastra Appendiculata Polysaccharides (CAPs)

In this study, the physicochemical properties of CAP were characterized, including its chemical composition, microscopic morphology, apparent viscosity, thermal properties, and triple helix structure. The results are summarized below and shown in Figure 1.
Chemical composition of CAP

The uronic acid content of CAP was determined to be 22.37 ± 0.20% using the sulfuric acid–carbazole method. The protein content was found to be 8.07 ± 0.25% using the Bradford method.
2.Microscopic morphology of CAP

The microscopic morphology of CAP at the 200×, 500×, and 1000× magnification is shown in Figure 1A. At the 200× magnification, CAP displayed an irregular lamellar and flaky morphology, with folds and scattered fragments of crude polysaccharides. At 500× magnification, the surface appeared slightly rough, and at 1000× magnification, flocculent crude polysaccharides were visible and distributed across the surface.
3.The apparent viscosity of CAP

The shear stress and apparent viscosity of CAP at different concentrations (1, 2, and 4 mg/mL) are shown in Figure 1B,C. At higher shear rates (10–100 S^−1^), significant differences in shear stress and apparent viscosity were observed (*p* < 0.05). Specifically, the 4 mg/mL concentration of CAP exhibited the highest shear stress and apparent viscosity, suggesting that a higher concentration positively influences the viscosity of CAP solutions.
4.Thermal properties of CAP

Thermal gravimetric analysis (TGA) was performed to assess the thermal stability of CAP by measuring the relationship between the sample mass and temperature change. As shown in Figure 1D, the weight–temperature variation curve for CAP showed two distinct weight loss stages. The first phase, occurring between 30 °C and 270 °C, involved a 9.9% mass loss, which likely corresponds to the loss of free water in CAP. The second significant stage of weight loss occurred between 270 °C and 335 °C, with an 82.38% mass reduction, likely due to the thermal decomposition of CAP and the breakdown of polymer chains. Additionally, the differential thermogravimetric (DTG) curve indicated that the maximum weight loss rate occurred at 314.43 °C. These results suggest that CAP remains relatively stable below 270 °C.
5.Identification of the triple helix structure of CAP

The triple helix structure of CAP was analyzed using Congo red dye, which forms complexes with polysaccharides in a triple helix conformation under weak alkaline conditions. The λ_max_ of the resulting complexes shifts to the red end of the spectrum compared to the Congo red reagent. However, as the NaOH concentration increases, the λ_max_ of the complexes decreases. As shown in Figure 1E, the λ_max_ of the Congo red reagent decreased with the increasing NaOH concentration. In contrast, the λ_max_ of the CAP–Congo red complex gradually increased within the NaOH concentration range of 0–0.3 mol/L, showing a significant redshift. This redshift was reversed at higher NaOH concentrations, indicating the presence of a triple helix structure in CAP.

**Figure 1 ijms-26-03900-f001:**
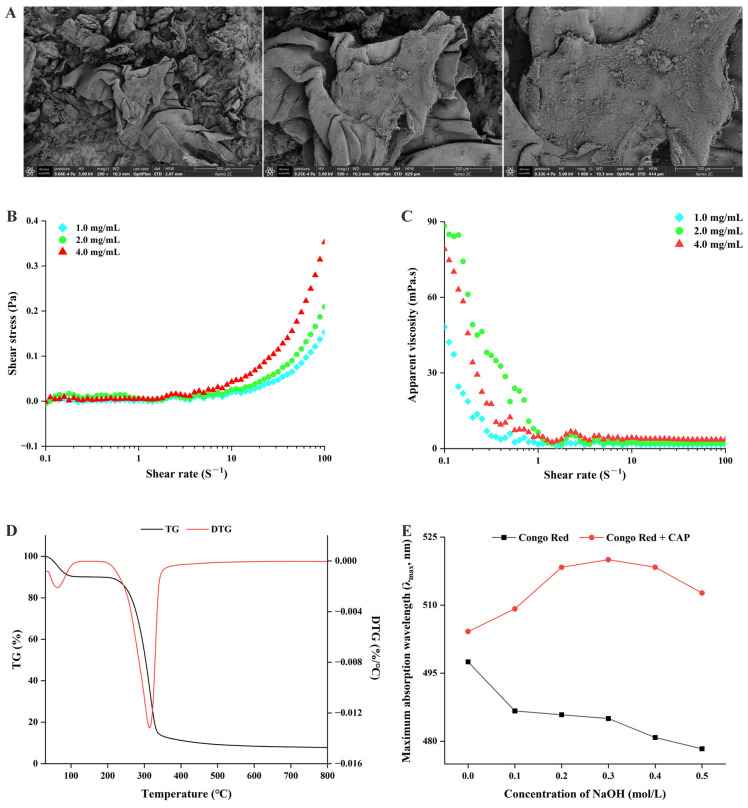
Physicochemical characterization of *Cremastra appendiculata* polysaccharides (CAPs). (**A**) Scanning electron micrographs at 200×, 500×, and 1000× resolutions; (**B**) shear stress of CAP solutions at different concentrations; (**C**) apparent viscosity of CAP solutions at different concentrations; (**D**) thermogravimetric analysis of CAP; and (**E**) Congo red test of CAP.

### 2.2. Evaluation of In Vitro Capacity of CAP

The antioxidant activity of CAPs was evaluated by measuring their free radical scavenging capacity, as shown in Figure 2A–C. CAP exhibited near-complete scavenging of hydroxyl radicals, with a rate approaching 100% (Figure 2A), which was significantly higher than its scavenging effects on ABTS·^+^ and DPPH radicals. The scavenging rates for ABTS·^+^ and DPPH radicals exhibited an increasing trend. At 2.0 mg/mL, CAP showed its highest scavenging rates of 49.10% and 39.29% for ABTS·^+^ and DPPH radicals, respectively (Figure 2B,C). These results demonstrate that CAP possesses strong antioxidant capacity in vitro.

The effect of CAP on Aβ_25–35_-treated rat adrenal pheochromocytoma (PC12) cells was used to assess its potential anti-AD activity. The results, presented in Figure 2D, show that compared with the control group, the survival rate of PC12 cells in the model group was significantly reduced by 38.24% after Aβ_25–35_ treatment (*p* < 0.001), indicating that Aβ_25–35_ caused damage to the cells. However, treatment with CAP significantly improved the survival rate of PC12 cells compared to the model group (*p* < 0.001). The highest survival rate, observed at a CAP concentration of 25 mg/mL, increased by 31.65% relative to the model group. These findings suggest that CAP may have a protective effect against Aβ_25–35_-induced damage in PC12 cells, indicating its potential to mitigate Alzheimer’s disease, warranting further investigation.

### 2.3. Effect of CAP on Tau-Induced Toxicity in Caenorhabditis elegans (C. elegans)

The effects of CAP on the physiological state of tau transgenic *C. elegans* (BR5270) were evaluated by measuring the lifespan, pharyngeal pump frequency, and body size. The survival curves of BR5270 shifted to the right following CAP treatment compared to the control group (Figure 3A). The mean and maximum lifespans increased with higher CAP concentrations. At 2.0 mg/mL, the mean lifespan of BR5270 increased by 16.73%, and the maximum lifespan was extended from 17 days in the control group to 22 days (Appendix A). The pharyngeal pump frequency of BR5270 did not change significantly (*p* > 0.05) after treatment with CAP, but the body size of BR5270 increased (*p* < 0.001) (Appendix A).

Due to tau protein aggregation in the body, BR5270 exhibits progressive locomotor defects. The swing frequency of BR5270 was measured on days 1, 3, 7, and 11 of CAP treatment. As shown in Figure 3B, the swing frequency of BR5270 decreased with age. On day 11, when the worms reached old age, CAP treatment at 2.0 mg/mL significantly increased the swing frequency by 36.19% (*p* < 0.01) compared to the control group. These results suggest that CAP can improve the locomotor defects in BR5270 worms caused by tau protein aggregation, demonstrating its potential to alleviate tau-induced toxicity.

### 2.4. Effect of CAP on Lifespan, Growth, Development, and Motility of Alzheimer’s Disease (AD) Model C. elegans

The survival curves of AD model CL4176 worms shifted to the right following CAP treatment compared to the control group (Figure 4A). The maximum lifespan of CL4176 increased in a concentration-dependent manner, extending from 25 days in the control group to 29 days at a CAP concentration of 2.0 mg/mL (Appendix A).

Pharyngeal pumping, body size, and swing are physiological indicators reflecting the health status of worms. The pharyngeal pump frequency and body size of AD model CL4176 worms did not change significantly (*p* > 0.05) after treatment with CAP (Appendix A).

The body swing frequency of CL4176 after CAP treatment is shown in Figure 4B. At CAP concentrations of 1.5 and 2.0 mg/mL, the swing frequency of CL4176 increased by 15.45% (*p* < 0.05) and 27.90% (*p* < 0.001), respectively, compared to the control group.

### 2.5. Effects of CAP on Amyloid-Beta (Aβ)-Induced Paralysis and 5-Hydroxytryptamine (5-HT) Hypersensitivity

Paralysis and 5-hydroxytryptamine (5-HT) hypersensitivity are two key AD phenotypes in *C. elegans* caused by Aβ deposition. In this study, the AD model worms CL4176 and CL2355 were used to assess the effect of CAP on paralysis and 5-HT hypersensitivity. The results are shown in Figure 5A,B and the Appendix A. CAP concentrations of 1.0, 1.5, and 2.0 mg/mL significantly extended the non-paralysis time of CL4176 in a concentration-dependent manner (Figure 5A). The best effect was observed at a concentration of 2.0 mg/mL, where the non-paralysis time was prolonged by 22.91% (*p* < 0.01) compared to the control group (Appendix A).

CL2355 worms, which exhibit rapid paralysis upon exposure to the exogenous 5-HT solution, were also tested. CAP at concentrations of 1.0, 1.5, and 2.0 mg/mL significantly reduced the rate of paralysis in CL2355 worms (*p* < 0.001) without affecting the control strain CL2122 (Figure 5B). The most pronounced reduction in 5-HT hypersensitivity was observed at 2.0 mg/mL, where the paralysis rate decreased by 41.18% compared to the control, bringing it to the level of the control strain. These findings suggest that CAP can alleviate both the paralysis and 5-HT hypersensitivity induced by Aβ deposition, indicating its potential to mitigate Aβ-induced toxicity.

The results also indicate that lower concentrations of CAP (0.1 and 0.5 mg/mL) had no significant effect on improving AD phenotypes in worms. Therefore, CAP concentrations of 1.0, 1.5, and 2.0 mg/mL were selected for subsequent experiments.

### 2.6. Effect of CAP on Aβ Deposition

In the transgenic *C. elegans* strain CL4176, Aβ_1–42_ deposition can be specifically stained with Thioflavin S. In this study, Thioflavin S staining was used to assess the impact of CAP on Aβ_1–42_ deposition. The results, shown in Figure 5C,D, demonstrate that CAP significantly reduced Aβ deposits in the heads of CL4176 worms (*p* < 0.001) in a concentration-dependent manner. At a CAP concentration of 2.0 mg/mL, the ratio of Aβ deposits to head area reached its lowest level, with a reduction of 79.96% compared to the control group. These findings indicate that CAP effectively reduces Aβ deposition in worms, suggesting that the observed improvements in paralysis and 5-HT hypersensitivity may be linked to this reduction in Aβ deposition.

### 2.7. Resistance Analysis

Oxidative stress is a key pathological feature and contributor to AD. Reducing oxidative stress can slow the progression of AD. In this study, the effect of CAP on the survival rates of AD model worms CL4176 and BR5270 under oxidative stress was examined. The results indicate that CAP significantly increases the survival rates of both worm strains under oxidative stress in a concentration-dependent manner (*p* < 0.05). At a concentration of 2.0 mg/mL, CAP had the most pronounced effect, extending the average lifespan of CL4176 by 257.41% (*p* < 0.001) and BR5270 by 43.05% (*p* < 0.001) compared to the control. The maximum lifespans were extended to 18 h and 11 h, respectively (Figure 6A,B, Appendix A).

Additionally, CAP enhanced the survival rate of BR5270 worms under heat stress in a concentration-dependent manner (*p* < 0.05). At a concentration of 2.0 mg/mL, the maximum lifespan of BR5270 under heat stress increased from 7 h in the control group to 9 h (Figure 6C, Appendix A), indicating that CAP improves resistance to heat stress in AD model worms.

### 2.8. Reactive Oxygen Species (ROS) Level and Antioxidant Enzyme Activities

Oxidative damage in worms can be assessed by measuring reactive oxygen species (ROS) levels. As shown in Figure 7A–D, CAP significantly reduced ROS levels in CL4176 worms in a concentration-dependent manner compared to the control (*p* < 0.001). At a CAP concentration of 2.0 mg/mL, the relative fluorescence intensity of ROS in CL4176 and BR5270 reached its lowest value, decreasing by 71.01% and 50.10%, respectively, compared to the control group. These results suggest that CAP can mitigate oxidative damage in AD model worms by scavenging ROS, demonstrating its in vivo antioxidant activity.

Glutathione S-transferase 4 (GST-4) and glutathione peroxidase (GSH-Px) are key antioxidant enzymes involved in metabolic detoxification, ROS scavenging, and maintaining redox homeostasis. To analyze the effect of CAP on the expression of GST-4, we used the transgenic *C. elegans* strain CL2166, which expresses a green fluorescent protein (GFP) gene. The results, shown in Figure 7E,F, indicate that CAP significantly increased GST-4 protein expression, with a 2.0 mg/mL concentration leading to an 84.75% increase in fluorescence intensity compared to the control (*p* < 0.001).

CAP also enhanced GSH-Px activity in the AD model worms CL4176 and BR5270 (Figure 7G,H). In both strains, GSH-Px activity increased in a concentration-dependent manner, reaching its highest level at 2.0 mg/mL. Compared to the control, GSH-Px activity in CL4176 increased by 289.52% (*p* < 0.001) and by 315.43% in BR5270 (*p* < 0.01). These findings suggest that CAP enhances the activities of antioxidant enzymes GST-4 and GSH-Px, supporting the hypothesis that CAP scavenges ROS and improves the antioxidant stress capacity of AD model worms by upregulating the activity of these enzymes.

### 2.9. Effect of CAP on Gene Expression in AD Model C. elegans

The transcription factors DAF-16 and SKN-1, associated with the insulin signaling pathway, regulate the expression of downstream antioxidant genes, playing key roles in aging and oxidative stress responses in worms. In this study, we analyzed the expression levels of *daf-16*, *skn-1*, and their downstream genes (*sod-3*, *ctl-1*, *gst-4*, and *hsp-70*) in the AD model worms CL4176 and BR5270 after treatment with CAP. The qRT-PCR results are shown in Figure 8.

In CL4176 worms, CAP treatment at 2.0 mg/mL significantly up-regulated the expressions of *skn-1*, *gst-4*, and *ctl-1* by 79.76%, 102.27%, and 177.29%, respectively (*p* < 0.001). The expression levels of *daf-16* and *hsp-70* were also significantly increased by 39.96% and 43.14%, respectively (*p* < 0.01), while *sod-3* showed a slight up-regulation of 21.02% (*p* < 0.05).

In BR5270 worms, CAP treatment led to significant up-regulation of *skn-1* (78.97%) and *gst-4* (68.44%) (*p* < 0.001). Additionally, *daf-16*, *hsp-70*, *ctl-1*, and *sod-3* were up-regulated by 64.26%, 57.09%, 70.09%, and 45.40%, respectively (*p* < 0.01). The upstream gene *daf-2*, which negatively regulates *daf-16* and *skn-1*, was significantly down-regulated by 29.58% (*p* < 0.001), while *hsf-1*, encoding heat shock transcription factors, was significantly up-regulated by 64.80% (*p* < 0.01).

These results suggest that CAP enhances the antioxidant capacity of AD model worms by up-regulating *daf-16* and *skn-1* in the insulin signaling pathway, along with their downstream antioxidant genes *ctl-1*, *sod-3*, *gst-4*, and *hsp-70*.

## 3. Discussion

*Cremastra appendiculata*, a traditional Chinese medicinal herb, has been used for thousands of years due to its various therapeutic properties, including antioxidant, anticancer, anti-inflammatory, and neuroprotective effects [25]. As a key component, CAPs have been shown to exhibit antioxidant, anti-tumor, and immunomodulatory activities [21,22,23]. The results of previous studies showed that CAP is rich in mannose and glucose, with a relative average molecular weight of 55.76 kDa, and contains functional groups such as O-H, C-H, C=O, and C-O [21]. In this study, the physicochemical properties of CAP were further characterized. CAP showed an irregular lamellar structure under a microscopic view and was thermally stable up to 270 °C, which is consistent with the results of Zhang et al. [23]. In addition, in this study, CAP contained 22.37% uronic acid. The results of the Congo red test and rheology show that CAP has a triple helix structure and a higher apparent viscosity at high concentrations (4 mg/mL). Studies have shown that the antioxidant activity of polysaccharides is proportional to their uronic acid content [26]. Yi et al. [27] showed that the uronic acid content of *Achyranthis bidentatae* radix polysaccharide increased from 4.32% to 6.77% after pretreatment with a steam explosion, and the scavenging capacity of this modified polysaccharide for free radicals also increased significantly. He et al. [28,29] showed that two polysaccharides, PUP80S1 and PUP60S2, extracted from *Polyporus umbellatus* sclerotia, had uronic acid contents of 8.5% and 22.3%, respectively. Compared with PUP80S1, PUP60S2, with a higher uronic acid content, had a higher scavenging rate of DPPH radicals. In this study, CAP contained 22.37% uronic acid, which is similar to PUP60S2. In addition, CAP can scavenge hydroxyl (·OH), ABTS·^+^, and DPPH radicals, and the scavenging rate of ·OH radicals is almost 100%, which may be related to the content of uronic acid in CAP.

Alzheimer’s disease (AD), the most common neurodegenerative disorder, has rapidly emerged as one of the most costly and deadly diseases of this century, with the number of AD patients worldwide projected to triple by 2050 [30]. Consequently, there has been significant interest in identifying drugs that can effectively alleviate AD symptoms. In recent years, polysaccharides from various natural sources have demonstrated anti-AD potential [17].

PC12 cells, derived from rat adrenal medullary pheochromocytoma, are commonly used as an in vitro model for AD research. Polysaccharides from *Codonopsis pilosula* improved cellular viability in an Aβ_1–40_-induced PC12 cell model [31]. In the current study, CAP similarly improved the survival rate of Aβ_25–35_-treated PC12 cells, suggesting that CAP possesses potential anti-AD activities in vitro. The protective mechanism of CAP against Aβ_25–35_-treated PC12 cells may be through downregulating the pro-apoptotic protein Bax, upregulating the anti-apoptotic protein Bcl-2, and inhibiting the activation of Caspase-3, thereby inhibiting the apoptotic signaling pathway [32,33].

To further explore the role and mechanism of CAP in alleviating AD, this study employed transgenic *C. elegans* strains CL4176 and BR5270 as in vivo models of AD. The CL4176 strain is transfected with the human Aβ_1–42_ gene, which is abundantly expressed in the body wall muscle when induced by elevated temperatures at 25 °C [34]. The BR5270 strain expresses the human tau protein, enabling the study of tau-related toxicity [35].

Previous studies have widely used these AD model worms to investigate AD pathology and potential treatments [14,15,16]. Key indicators of worm health include body size, pharyngeal pumping, locomotion, and lifespan. Natural products with anti-AD activity can improve these health markers to varying degrees. For example, Li et al. reported that the *Coptis chinensis* Franch polysaccharide at 100 mg/L significantly extended the lifespan of the AD model worm CL4176 [36]. Similarly, Xiao et al. demonstrated that durian seed polysaccharides at concentrations of 1, 3, and 5 mg/mL significantly prolonged the lifespan of CL4176 worms [37]. Compared with Xiao et al.’s study [37], CAP significantly prolonged the lifespan of CL4176 worms at lower concentration ranges (0.1, 0.5, 1.0, 1.5, and 2.0 mg/mL), suggesting that CAP is more effective in prolonging lifespan relative to durian seed polysaccharides.

Transgenic *C. elegans* BR5270 has a shorter lifespan than wild-type worms due to the toxic aggregation of tau proteins [38]. In this study, CAP significantly extended the lifespan and promoted the growth of BR5270 worms. At a concentration of 2.0 mg/mL, the swing frequencies of CL4176 and BR5270 increased by 27.90% and 36.19%, respectively. These findings suggest that CAP not only prolongs lifespan but also enhances the locomotor ability of AD model *C. elegans* without impairing normal growth. Therefore, CAP appears to improve the overall health status of AD model worms.

Brain amyloid and cerebrospinal fluid (CSF) tau are clinical biomarkers of AD [10]. The detection of abnormal proteins in CSF, including β-amyloid and phosphorylated tau, can be utilized for the clinical diagnosis and monitoring of neurodegenerative diseases [6]. Therefore, we assessed the effects of CAP on Aβ and tau-related pathology in AD model worms. Following temperature induction at 25 °C, the expression of Aβ_1–42_ in the body wall muscles of the transgenic *C. elegans* strain CL4176 resulted in a paralyzed phenotype, characterized by a rigid, immobile body only showing head movement [25]. In the CL2355 strain, Aβ_1–42_ expression in neurons leads to hypersensitivity to 5-HT, causing paralysis [39]. In this study, treatment with 2 mg/mL CAP prolonged the non-paralysis time of CL4176 by 22.91% and reduced the paralysis rate of CL2355 by 41.18%. Notably, at a lower concentration of 0.1 mg/mL, CAP treatment achieved a 60% non-paralyzed rate in CL4176 (thermally induced at 25 °C for 36 h), which is better than that of *Cibotium barometz* polysaccharide (non-paralysis rate < 60%) [19]. These results demonstrate that CAP exhibited potent inhibitory effects on Aβ-induced toxicity. To further investigate the molecular mechanisms of CAP-mediated paralysis inhibition, we assessed Aβ deposition in the head region of CL4176 worms using Thioflavin staining. Treatment with 2 mg/mL CAP resulted in a 79.97% reduction in the ratio of Aβ deposits in the head area of CL4176, demonstrating superior efficacy compared to *Amanita caesarea* polysaccharides, which showed a 26.2% decrease in Aβ_1–42_ levels [40]. These results suggest that CAP exerts neuroprotective effects by reducing Aβ deposition and delaying the paralysis phenotype in CL4176.

Tau protein aggregation in the BR5270 worm results in progressive locomotor defects [34]. Additionally, heat induction has been shown to accelerate tau phosphorylation, and mitigating heat stress can reduce tau hyperphosphorylation [41]. In this study, CAP at a concentration of 2.0 mg/mL increased the swing frequencies of BR5270 by 36.19% and significantly enhanced the survival rate of BR5270 under heat stress. These findings suggest that CAP can alleviate tau aggregation-induced toxicity, confirming its neuroprotective activity. We hypothesize that the mechanism of CAP inhibiting tau toxicity may involve the suppression of glycogen synthase kinase 3β (GSK3β) activity through inhibiting phosphorylation at the site of tyrosine 216 and increasing phosphorylation at the site of serine 9, thereby inhibiting tau hyperphosphorylation [42]. Alternatively, polysaccharides might directly interfere with tau aggregation and fibril formation by establishing hydrogen bonds or hydrophobic interactions with specific amino acid residues in the tau protein sequence [43,44].

In this study, CAP demonstrated the significant alleviation of both Aβ- and Tau-related pathologies in AD model worms, indicating its potential as a therapeutic candidate for AD.

Oxidative stress plays a critical role in AD progression, especially in the early stages, where increased oxidative stress can enhance the activity of β- and γ-secretases, leading to the overproduction of Aβ [45,46]. Excessive Aβ exacerbates mitochondrial dysfunction, promotes inflammation, and induces the production of ROS, further aggravating oxidative damage [47]. Therefore, reducing oxidative stress is a key therapeutic strategy for AD [48]. Previous studies have shown that natural compounds such as Rehmanniae Radix oligosaccharides [49] and deer antler extracts [50] alleviate Aβ-induced paralysis and reduce ROS production by enhancing antioxidant enzyme activities. In this study, CAP similarly improved the survival of CL4176 and BR5270 worms under oxidative stress, increased GSH-Px activity, upregulated GST-4 expression, and scavenged ROS in vivo, which proves the antioxidant activity of CAP in vivo.

To further investigate the mechanism by which CAP alleviates AD symptoms, we analyzed related signaling pathways. Previous research has shown that CAP promotes the nuclear localization of DAF-16, activating downstream antioxidant genes such as *sod-3*, *ctl-1*, and *hsp-16.2*, thereby enhancing stress resistance and prolonging the lifespan of wild-type *C. elegans* [21]. DAF-16, a homolog of mammalian FOXO, is a key regulator in the insulin/IGF-1 signaling pathway and has been shown to modulate the formation of less toxic, high-molecular-weight Aβ aggregates [51]. DAF-16-regulated genes *sod-3* and *ctl-1* encode manganese superoxide dismutase and catalase, both of which are essential for maintaining redox balance. Additionally, SKN-1, a homolog of mammalian Nrf2, plays a crucial role in longevity and oxidative stress resistance and has been shown to extend the lifespan and mitigate Aβ toxicity in *C. elegans* [52,53]. The SKN-1-regulated gene *gst-4* encodes glutathione S-transferase, which is implicated in many neurodegenerative diseases, with reduced activity observed in AD patients [54]. Moreover, *hsp-70*, which encodes heat shock proteins, plays a vital role in preventing protein misfolding, which is a common feature of neurodegenerative diseases [35]. Based on this, we hypothesized that CAP could alleviate AD through the DAF-16 signaling pathway.

To verify this hypothesis, we examined the mRNA expression levels of the above genes in AD model worms CL4176 and BR5270. The qRT-PCR results revealed the significant upregulation of *daf-16*, *skn-1*, and their target genes following CAP treatment. These findings suggest that CAP alleviates Aβ_1–42_ and tau-induced toxicity in AD model worms by enhancing antioxidant capacity through the activation of DAF-16 and SKN-1 signaling pathways.

## 4. Materials and Methods

### 4.1. Materials

*Cremastra appendiculata* was purchased in 2023 from Beijing Tongrentang (Wandong Road Store, Chengdu, China), and produced in Sichuan Province. Rat adrenal pheochromocytoma (PC12) cells were provided by Professor Yina Huang, Sichuan University, China. The amyloid-beta peptide fragment (25–35) (Aβ_25–35_, catalog number HY-P0128) was purchased from Shanghai MCE Company (Shanghai, China). Transgenic *C. elegans* strains CL2355 *{dvIs50 [pCL45 (snb-1::Aβ 1-42::3ʹ UTR (long) + mtl-2::GFP] I}*, CL2122 *{dvIs15 [(pPD30.38) unc-54 (vector) + (pCL26) mtl-2::GFP]}*, CL2166 *{dvIs19 [(pAF15)gst-4p::GFP::NLS] III}*, and uracil-deficient *Escherichia coli* (*E. coli* OP50) were provided by Professor Yuan Wang, State Key Laboratory of Biotherapy, Sichuan University. The transgenic *C. elegans* strain CL4176 *{dvIs27 [myo-3p::Aβ (1–42)::let-851 3ʹ UTR) + rol-6 (su1006)] X}* was provided by Professor Ming Yuan, Sichuan Agricultural University, and strain BR5270 *{byIs161 [rab-3p::F3(ΔK280) + myo-2p::mCherry]}* was provided by Professor Lingjun Zheng, Shanghai Jiao Tong University.

### 4.2. Extraction, Isolation, and Physicochemical Characterization of CAPs

CAPs were extracted using water extraction followed by alcohol precipitation, protein removal with a sevage reagent (n-butanol/trichloromethane = 1:4, *V*/*V*), and vacuum freeze-drying, as described by Wang et al. [21]. The physicochemical properties of CAP were characterized through the analysis of its chemical composition, micromorphology, apparent viscosity, thermal properties, and triple helix structure.

Protein Content: Protein content was determined using the Bradford method, with bovine serum albumin used as the standard, following the method described by Abubakar et al. [55] with modifications.

Uronic Acid Content: The uronic acid content was measured using the sulfuric acid–carbazole method with D-glucuronic acid as the standard, following the method described by Abubakar et al. [55] with modifications.

Microscopic Morphology: Scanning electron microscopy (SEM, Thermo Scientific Apreo 2C, Thermo Fisher Scientific, Waltham, MA, USA) was employed to observe the microstructure of CAPs. A small amount of freeze-dried CAP was spread onto a conductive adhesive, coated with gold by ion sputtering, and then examined at magnifications of 200×, 500×, and 1000× under a 15 kV voltage [23].

Apparent Viscosity: The apparent viscosities of CAP solutions (1, 2, and 4 mg/mL) were measured using a rotational rheometer (MCR302, Anton Paar, Graz, Austria) at 25 °C across shear rates of 0.1 to 100 S^−1^ [56].

Thermal Properties: Thermal gravimetric analysis (TGA) was performed using a Mettler Toledo TGA2 (Mettler Toledo, Zurich, Switzerland) under a nitrogen atmosphere. The scanning temperature range was set from 30 °C to 700 °C with a heating rate of 10 °C/min [23].

Triple Helix Structure: The triple helix structure of CAP was analyzed using Congo red dye, following modifications of the method by Liu et al. [57]. In a 10 mL centrifuge tube, 1.5 mL of 2 mg/mL CAP was mixed with 1.5 mL of 160 μg/mL Congo red solution. The reaction was adjusted to final NaOH concentrations of 0, 0.1, 0.2, 0.3, 0.4, and 0.5 mol/L. After incubating at room temperature for 10 min, the maximum absorption wavelengths (λ_max_) of the reaction solutions were measured within the range of 400–600 nm. Solutions containing only Congo red and NaOH were used as controls.

### 4.3. Determination of In Vitro Antioxidant Capacity of CAPs

The antioxidant activities of CAPs were assessed using the hydroxyl radical (·OH), ABTS·^+^, and DPPH radical scavenging assays, with minor modifications from previously reported methods [22,58].
(1)Hydroxyl radical (·OH) scavenging activity

CAP was dissolved in distilled water (ddH_2_O) to prepare solutions at concentrations of 0, 0.1, 0.5, 1.0, 1.5, and 2.0 mg/mL. Hydroxyl radicals were generated by mixing 200 μL of 9 mM FeSO_4_ and 200 μL of 10 mM H_2_O_2_ in a 2 mL test tube. Immediately, 200 μL of the CAP solution was added, followed by 200 μL of 9 mM salicylic acid. The mixture was incubated at 37 °C for 30 min. Absorbance was measured at 510 nm using a spectrophotometer. L-ascorbic acid (vitamin C, VC) at equivalent concentrations was used as a positive control. The hydroxyl radical scavenging rate was calculated using the following formula:Scavenging rate (%) = [(1 − (*A*_1_ − *A*_0_)/*A*_2_)] × 100,
where *A*_1_ is the absorbance of the solution containing CAP or VC, *A*_0_ is the absorbance of the sample itself, and *A*_2_ is the absorbance of the blank tube.
(2)ABTS·^+^ radical scavenging activity

ABTS·^+^ radicals were generated by reacting 7 mM ABTS with 4.9 mM potassium persulfate in the dark at room temperature for 16 h. The resulting ABTS·^+^ solution was diluted with phosphate buffer (pH 7.4) to an absorbance of 0.7 ± 0.02 at 734 nm. CAP solutions were prepared at concentrations of 0 to 2.0 mg/mL, as shown above. A mixture of the 0.1 mL CAP solution and 3.9 mL ABTS·^+^ solution was incubated at room temperature for 6 min. Absorbance was then measured at 734 nm. L-ascorbic acid served as the positive control. The ABTS·^+^ radical scavenging rate was calculated as follows:Scavenging rate (%) = [(1 − *A*_1_/*A*_0_)] × 100,
where *A*_1_ is the absorbance of the ABTS·^+^ solution containing CAP or VC, and *A*_0_ is the absorbance of the blank solution.
(3)DPPH radical scavenging activity

CAP solutions (0 to 2.0 mg/mL) were prepared as described. In each test tube, 630 μL of the 0.2 mM DPPH solution and 70 μL of CAP solution were mixed thoroughly. The mixture was incubated at 37 °C for 30 min. Absorbance was measured at 515 nm. L-ascorbic acid was used as the positive control. The DPPH radical scavenging rate was determined using the following equation:Scavenging rate (%) = [(1 − (*A*_1_ − *A*_0_)/*A*_2_)] × 100,
where *A*_1_ is the absorbance of the reaction solution containing CAP or VC, *A*_0_ is the absorbance of the sample itself, and *A*_2_ is the absorbance of the blank tube.

### 4.4. Determination of In Vitro Anti-AD Capacity of CAP

The viability of PC12 cells was assessed using a Cell Counting Kit-8 (CCK-8, BS350A, Biosharp, Hefei, China), following the method described by Hu et al. [59] with modifications. PC12 cells in the logarithmic growth phase were harvested using trypsin digestion, centrifuged to remove the supernatant, and resuspended in a complete medium. Cells were counted using a hemocytometer under a microscope and diluted to a concentration of 1 × 10^5^ cells/mL. They were then seeded into 96-well plates at 100 μL per well (1 × 10^4^ cells/well) and incubated at 37 °C with 5% CO_2_ for 24 h.

The cells were then divided into three groups: control, model, and CAP-treated groups. The control group was incubated in a culture medium without Aβ_25–35_ or CAP for 96 h. The model group was exposed to 200 μM Aβ_25–35_ in the culture medium for 48 h. The CAP-treated group was first incubated with different concentrations of CAP (15, 20, 25 mg/mL) for 48 h, followed by exposure to 200 μM Aβ_25–35_ for an additional 48 h. After treatment, 10 μL of the CCK-8 reagent was added to each well and incubated for 2 h. Absorbance at 450 nm was then measured using a microplate reader to determine cell viability.

### 4.5. C. elegans Strain and Experimental Design

All *C. elegans* strains were cultured on nematode growth medium (NGM) plates seeded with live *E. coli* OP50 as a food source. The transgenic strains CL4176, CL2355, CL2122, and BR5270 were maintained at 16 °C, while CL2166 was cultured at 20 °C. To synchronize the worms, eggs were obtained using the sodium hypochlorite method as described by Porta-de-la-Riva et al. [60], ensuring all worms were at the same developmental stage. The eggs were incubated on NGM plates seeded with live *E. coli* OP50 until they reached the L4 larval stage. At this point, worms were transferred to fresh NGM plates that were pre-seeded with live *E. coli* OP50 bacterial lawns and overlaid with different concentrations of CAP (0, 0.1, 0.5, 1.0, 1.5, and 2.0 mg/mL) after air-drying for further experiments.

### 4.6. Determination of Physiological Indexes


(1)Lifespan Assay:


Fifty synchronized L4 larvae were transferred to NGM plates with varying CAP concentrations and incubated at 16 °C. The worms were observed every 24 h, starting from day 0, and then transferred to a new NGM plate to exclude the influence of offspring on the experiment. Survival was assessed daily by gently touching the head and tail with a platinum wire; lack of movement indicated death. Observations continued until all the worms had died [61].
(2)Pharyngeal Pumping and Body Size:

Approximately thirty synchronized L4 larvae per group were transferred to NGM plates with varying CAP concentrations and cultured at 16 °C for 4 days. Then, six worms from each group were randomly selected. For pharyngeal pumping rates, worms were placed on slides with 2% agar and a small amount of live *E. coli* OP50; the number of pharyngeal contractions within 1 min was recorded under a microscope. For body size measurements, worms were photographed using Image Pro Plus 6.0 software [DS-Ri1-U3 (ver=1010.0100.20.00.0201.0208.01000000], and body length and width were measured using ImageJ software [Java 1.8.0_345 (64-bit)] [21].
(3)Motility Assay:

Approximately thirty synchronized L4 larvae per group were transferred to NGM plates with varying CAP concentrations and cultured at 16 °C for 4 days. After 4 days of CAP treatment, six CL4176 worms from each group were transferred to a 24-well plate containing 1 mL of the M9 buffer. The number of complete sinusoidal movements (swings) within 30 s was recorded under a microscope. For BR5270 worms, motility was assessed on days 1, 3, 7, and 11 of CAP treatment using the same method [21].

### 4.7. Stress Resistance Assays

Approximately thirty synchronized L4 larvae were transferred to NGM plates treated with varying concentrations of CAP and cultured at 16 °C for 4 days. After treatment, 15 worms from each group were randomly selected for acute exposure to either 1% H_2_O_2_ solution on NGM plates or a high temperature (37 °C). The number of deaths and survivors was recorded every hour until all worms had died [21].

### 4.8. Paralysis, 5-Hydroxytryptamine Hypersensitivity Assays

Synchronized eggs from the transgenic *C. elegans* strain CL4176 were transferred to NGM plates and incubated at 16 °C for 48 h. Fifteen hatched larvae from each group were randomly selected and moved to NGM plates containing various concentrations of CAP. The worms were then cultured at 25 °C for an additional 34 h. Paralysis was assessed every two hours by gently touching the worms with a platinum wire; worms were considered paralyzed if they exhibited stiffness and immobility, with movement only in the head region [14].

For the 5-hydroxytryptamine (5-HT) hypersensitivity assay, synchronized eggs from transgenic strains CL2355 and CL2122 were cultured at 16 °C until they reached the L1 larval stage. Fifteen larvae from each group were randomly selected and transferred to NGM plates with varying concentrations of CAP. After 96 h of culture at 16 °C, the worms were incubated at 25 °C for an additional 36 h. Post-incubation, worms were rinsed with the M9 buffer and transferred to a 24-well plate containing 1 mL of 5 mg/mL 5-HT solution. The number of paralyzed worms was recorded after 5 min of exposure [62].

### 4.9. Aβ Deposition

Synchronized eggs from CL4176 worms were incubated on NGM plates at 16 °C for 48 h. Approximately thirty hatched worms were then transferred to NGM plates containing different concentrations of CAP and cultured at 25 °C for 34 h. After incubation, worms were rinsed with the M9 buffer to remove residual *E. coli* OP50 and fixed in 1 mL of 4% paraformaldehyde at 4 °C for 24 h. Fixed worms were permeabilized with 1 mL of solution containing 1% Triton X-100, 5% β-mercaptoethanol, and 125 mM Tris (pH 7.4) at 37 °C for 24 h. Following three washes with the M9 buffer, worms were stained with 1 mL of 0.125% Thioflavin S solution (dissolved in 50% ethanol) for 2 min. Any excess stain was removed by washing with 50% ethanol three times for 5 min each. At least six worms per group were mounted on slides containing 2% agar, visualized under a fluorescence microscope, and images were analyzed using ImageJ software [62]. Aβ deposition was quantified by calculating the ratio of the fluorescent area to the total head area of the worm.

### 4.10. Determination of Intracellular ROS and Antioxidant Enzyme Activities

Reactive oxygen species (ROS) levels were measured using the cell-permeable fluorescent probe DCFH-DA (S0033S, Beyotime, Shanghai, China), following the method described by Yang et al. [63] with slight modifications. Approximately thirty synchronized L4 larvae were transferred to NGM plates and treated with different concentrations of CAP and cultured at 25 °C for 36 h (CL4176) or at 16 °C for 7 days (BR5270, with a medium change to fresh CAP-containing NGM plates on day 4 of the treatment). Afterward, the worms were rinsed three times with the M9 buffer to remove *E. coli* OP50 from their surfaces. The worms were then incubated in a DCFH-DA solution at a final concentration of 100 μM for 3 h at 25 °C (CL4176) or 16 °C (BR5270) in the dark. Finally, the worms were transferred to slides containing 2% agar and photographed under a fluorescence microscope with excitation and emission wavelengths of 485 nm and 530 nm, respectively. At least six worms were imaged per group, and the fluorescence intensity was analyzed using ImageJ software.

The in vivo expression of the glutathione S-transferase 4 (GST-4) protein was measured using transgenic *C. elegans* strain CL2166 *{dvIs19 [(pAF15)gst-4p::GFP::NLS] III}*. Approximately thirty synchronized L4 larvae were transferred to NGM plates treated with varying concentrations of CAP and cultured at 20 °C for 4 days. Six worms from each group were randomly selected, placed on 2% agar slides, and anesthetized with 0.5% of the NaN_3_ solution. The imaging method for GFP-carrying transgenic worms under a fluorescence microscope was performed as described in previous studies [21]. The worms were then observed and photographed using a fluorescence microscope, with excitation and emission wavelengths of 485 nm and 530 nm, respectively, and the fluorescence intensity was analyzed using ImageJ software.

Glutathione peroxidase (GSH-Px) activity was assessed according to the kit instructions. Approximately four hundred synchronized L4 larvae were transferred to NGM plates treated with different concentrations of CAP and cultured at 16 °C for 4 days. The worms were rinsed with the M9 buffer to remove *E. coli* OP50 and then lysed on ice using a cell crusher. The supernatants were collected by centrifugation (4000 rpm for 20 min) and analyzed using a glutathione peroxidase assay kit (A005-1-2, Nanjing Jiancheng Bioengineering Institute, Nanjing, China) and a BCA protein concentration assay kit (P0012, Beyotime, Shanghai, China) to determine GSH-Px activity and the total protein content.

### 4.11. Real-Time Quantitative Polymerase Chain Reaction

Approximately four hundred synchronized L4 larvae were transferred to NGM plates, treated with different concentrations of CAP (0 and 2.0 mg/mL), and cultured at 25 °C for 36 h (CL4176) or at 16 °C for 4 days (BR5270). The total RNA was extracted from the worms using an RNA extraction kit (BL1179, Yeasen, Shanghai, China), following the manufacturer’s instructions. The extracted RNA was then reverse-transcribed into complementary DNA (cDNA). SYBR Green was used for detection, with act-1 serving as the internal reference gene. Primers were designed using the NCBI Primer Design Tool and synthesized by Tsingke Biotechnology Co., Ltd., Beijing, China. The specific primer sequences are listed in the Appendix A.

### 4.12. Statistical Analysis

All experiments were conducted in triplicate. Data were analyzed using SPSS version 25.0 (IBM Corp., Armonk, NY, USA) employing one-way analysis of variance (ANOVA) followed by least significant difference (LSD) post hoc tests. Survival data from lifespan assays were analyzed using the Kaplan–Meier method. Relative gene expression levels were calculated using the 2^−ΔΔCT^ method. Data visualization was performed using GraphPad Prism version 9 (GraphPad Software, San Diego, CA, USA) and Microsoft Excel 2021.

## 5. Conclusions

CAP has a triple helix structure and is stable below 270 °C. Under a scanning electron microscope, CAP shows an irregular layered structure. In terms of biological activity, CAP demonstrates significant antioxidant and anti-AD activity. In vitro, CAP exhibits both antioxidant properties and the ability to mitigate AD-related effects. In in vivo models using the AD transgenic *C. elegans* strains CL4176 and BR5270, CAP effectively alleviates the motor dysfunction caused by tau protein aggregation, reduces paralysis and hypersensitivity to 5-HT caused by Aβ, and decreases Aβ deposition. Moreover, CAP scavenges ROS, enhances the activity of key antioxidant enzymes like GST-4 and GSH-Px, and upregulates the expression of critical genes (*daf-16*, *skn-1*, *gst-4*, *ctl-1*, *sod-3*, and *hsp-70*) via the insulin signaling pathway. This gene regulation contributes to the improved antioxidant capacity of AD model *C. elegans*, highlighting CAP’s in vivo antioxidant activity. CAP exerts its anti-AD effects by reducing the toxicity induced by Aβ and tau proteins while mitigating oxidative stress. These findings support the potential of CAP as a promising therapeutic candidate for Alzheimer’s disease.

## Figures and Tables

**Figure 2 ijms-26-03900-f002:**
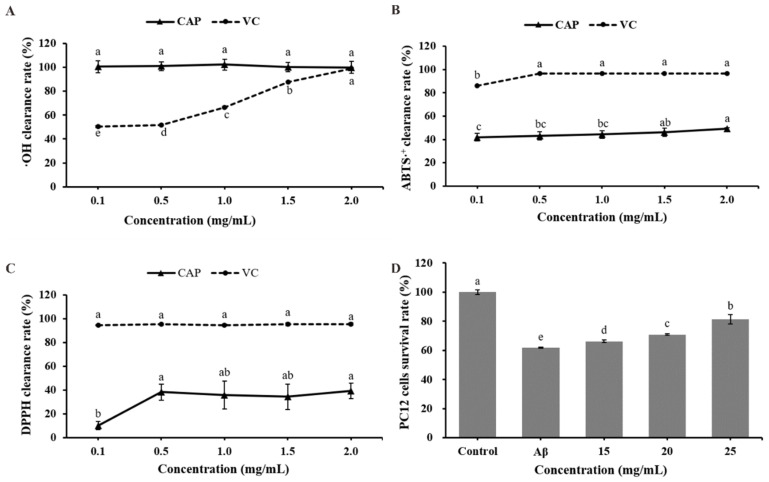
In vitro antioxidant and anti- Alzheimer’s disease (AD) activities of CAP. (**A**) hydroxyl (·OH) radical scavenging rate; (**B**) ABTS·^+^ radical scavenging rate; (**C**) DPPH radical scavenging rate; and (**D**) the viability of amyloid-beta peptide fragment (25–35) (Aβ_25–35_)-treated PC12 cells. Bars with different letters are significantly different (*p* < 0.05).

**Figure 3 ijms-26-03900-f003:**
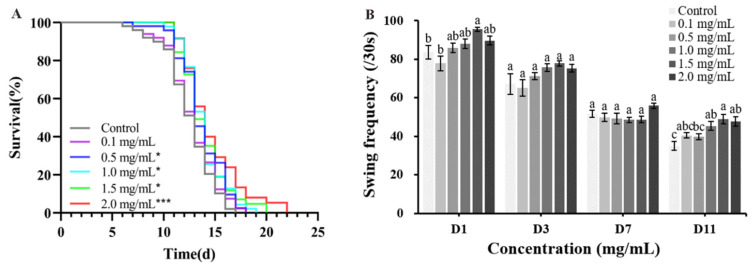
The effects of CAP on Tau-induced toxicity in *Caenorhabditis elegans* (*C. elegans*). (**A**) Survival curves of BR5270. Significant differences relative to the control (* *p* < 0.05 and *** *p* < 0.001) are shown. (**B**) The swing frequency of BR5270 in 30 s. Bars with different letters are significantly different (*p* < 0.05).

**Figure 4 ijms-26-03900-f004:**
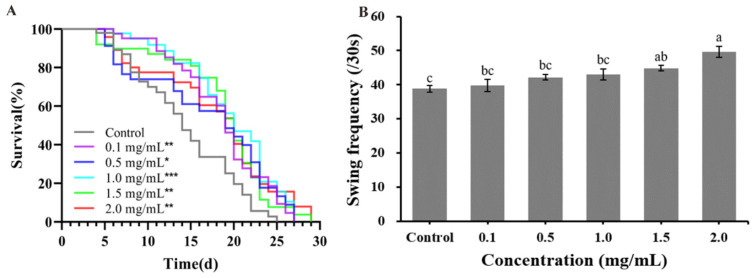
The effects of CAP on the lifespan and motility of AD model *C. elegans*. (**A**) The survival curves of CL4176. Significant differences relative to the control (* *p* < 0.05, ** *p* < 0.01 and *** *p* < 0.001) are shown. (**B**) The swing frequency of CL4176 after 30 s. Bars with different letters are significantly different (*p* < 0.05).

**Figure 5 ijms-26-03900-f005:**
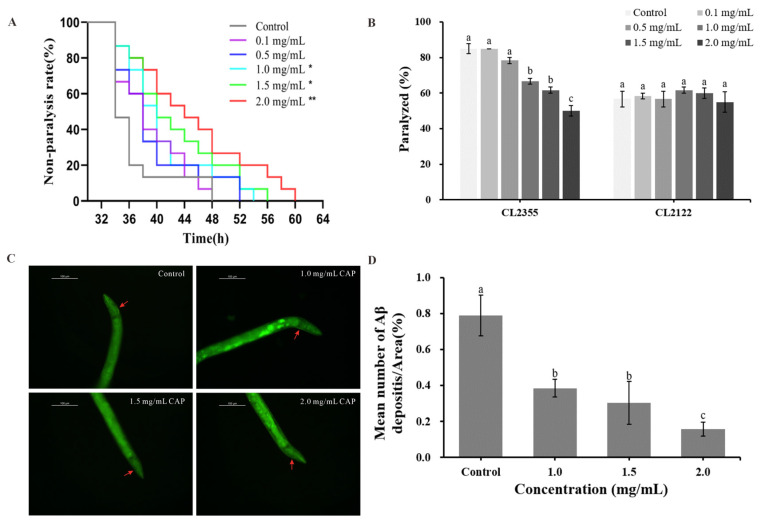
The effects of CAP on paralysis, 5-hydroxytryptamine (5-HT) hypersensitivity, and amyloid-beta (Aβ) deposition in AD model *C. elegans*. (**A**) Paralysis of CL4176; (**B**) hypersensitivity of CL2355 and CL2122 to 5-HT; (**C**) Aβ deposition in CL4176. Aβ aggregates are remarked with red arrows; and (**D**) the quantitative data of Aβ deposition in CL4176. Significant differences relative to the control (* *p* < 0.05 and ** *p* < 0.01) are shown. Bars with different letters are significantly different (*p* < 0.05).

**Figure 6 ijms-26-03900-f006:**
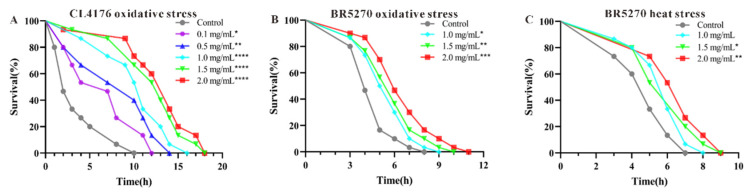
The effects of CAP on the stress resistance of AD model *C. elegans*. (**A**) Oxidative stress in CL4176; (**B**) oxidative stress in BR5270; (**C**) heat stress in BR5270. Significant differences relative to the control (* *p* < 0.05, ** *p* < 0.01, *** *p* < 0.001 and **** *p* < 0.0001) are shown.

**Figure 7 ijms-26-03900-f007:**
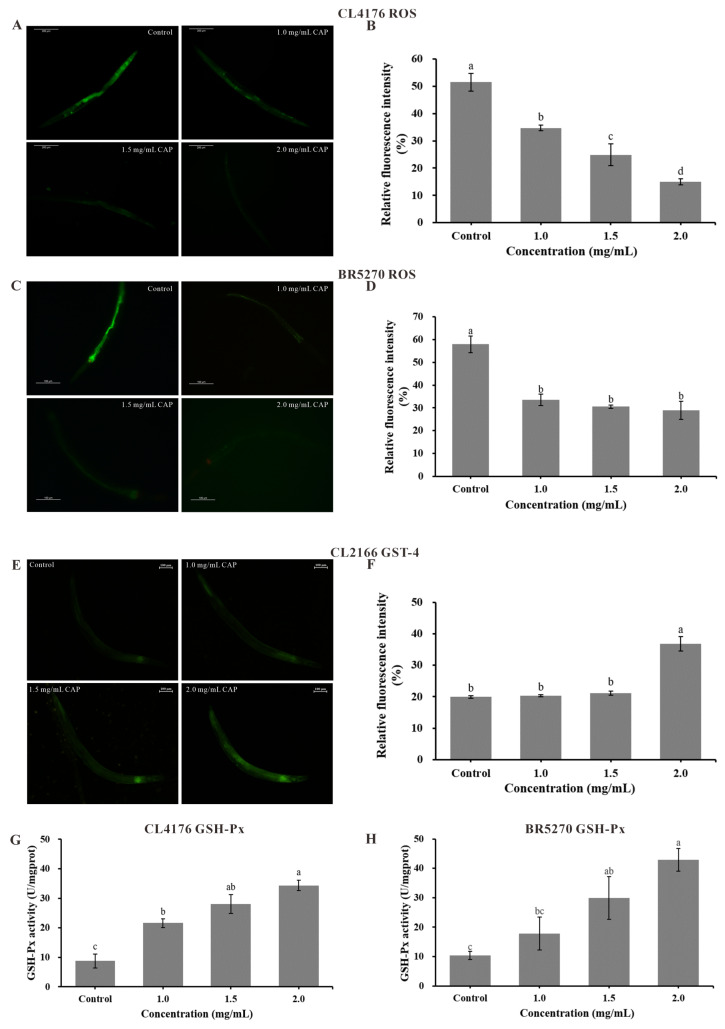
The effects of CAP on reactive oxygen species (ROS) levels and antioxidant enzyme activities in AD model *C. elegans*. The accumulation of ROS in CL4176 (**A**) and BR5270 (**C**); quantitative data corresponding to ROS accumulation in CL4176 (**B**) and BR5270 (**D**); (**E**) the accumulation of GST-4::GFP in CL2355; (**F**) quantitative data corresponding to GST-4::GFP accumulation in CL2355; and the GSH-Px enzyme activity in CL4176 (**G**) and BR5270 (**H**). Bars with different letters are significantly different (*p* < 0.05).

**Figure 8 ijms-26-03900-f008:**
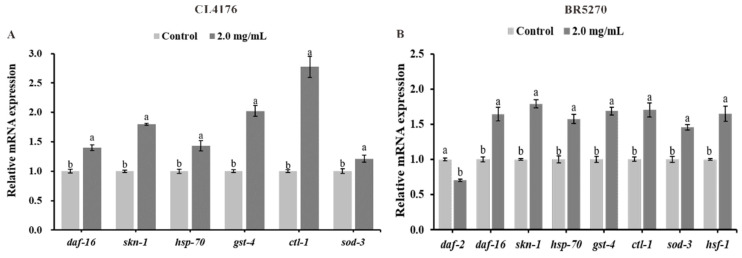
The effects of CAP on gene expression in AD model *C. elegans* CL4176 (**A**) and BR5270 (**B**). Bars with different letters are significantly different (*p* < 0.05).

## Data Availability

The data will be made available upon request.

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
