# Peer review of "Cremastra appendiculata Polysaccharides Alleviate Neurodegenerative Diseases in Caenorhabditis elegans: Targeting Amyloid-β Toxicity, Tau Toxicity and Oxidative Stress"

_ijms, 2025, doi:10.3390/ijms26083900_

Round 1
Reviewer 1 Report
Comments and Suggestions for Authors
The main objective of this study is to evaluate the effect of Cremastra appendiculata polysaccharides on a model of Alzheimer's disease in C. elegans. The same authors have already published an article with the same polysaccharides showing an effect on longevity and resistance to stress in C. elegans. doi: 10.1016/j.ijbiomac.2022.12.234. In general, however, research into this polysaccharide is still in its early stages, as there have been few studies published.
Below are my considerations for improving the work:
1. Abbreviations. I suggest putting them in written form and not in table format.
2. I detected a high percentage of plagiarism in the work, around 61% according to plagiarism checkers. I strongly suggest restructuring the work.
3. I suggest restructuring the title of the work. The paper evaluated several pathways related to Parkinsonism and Alzheimer's. I suggest making the title broader for neurodegenerative diseases.
4. As with the title, I suggest making the introduction and discussion more comprehensive for neurodegenerative diseases.
5. Several references are missing from the methodology. Please include the references for all the methods cited.
6. Make it clear in the methodology how the worms were treated with the polysaccharide: how was the polysaccharide placed on the plate with the C. elegans? Under the NGM medium? was the bacterium alive or dead? was the treatment done from which larval stage? was the treated medium renewed daily? number of worms per treatment and number of independent experiments.
7. The results are well described. I suggest posting images of the experiments with fluorescent worms. I suggest making the results with strains related to AD and Tau clear and separate.
In general, the work should be reviewed for the high percentage of plagiarism and the results should be treated more comprehensively as neurodegenerative diseases. The methodology should be better described. The text as a whole should be revised to improve fluidity and readability.
Reviewer 2 Report
Comments and Suggestions for Authors
The manuscript by Xu et al. investigates the potential of Cremastra appendiculata polysaccharides (CAP) as a neuroprotective agent in an Alzheimer's disease model. This study contributes to the growing body of research on plant-derived compounds and their potential applications in developing drugs and supplements for the prevention or treatment of Alzheimer's disease. Cremastra appendiculata, previously reported for its neuroprotective properties, is further explored here, adding valuable insights into its therapeutic potential.
Overall, this is a commendable piece of work. However, one notable limitation is the partial chemical characterization of the CAP mixture. To strengthen the manuscript, I recommend a complete chemical characterization of the mixture, identifying its individual components. This would provide a clearer understanding of the mechanisms underlying CAP's neuroprotective effects and enhance the study's scientific rigor.
The authors convincingly demonstrate that CAP is a promising candidate for developing Alzheimer's-related products based on their experimental findings. Nevertheless, comprehensive characterization remains essential to fully validate its potential.
On a positive note, the references cited are relevant, and the figures are well-presented, effectively supporting the study's findings.
Reviewer 3 Report
Comments and Suggestions for Authors
The manuscript is generally well written and presents comprehensive findings. However, more in-depth discussion is required regarding the differential effects and potential of Cremastra appendiculata polysaccharides (CAP) in Alzheimer's disease (AD) models.
- In the Introduction, while the antioxidant, anti-aging, anti-tumor, and immunomodulatory properties of CAP are mentioned, prior studies specifically addressing the effects of CAP on AD should be introduced to establish a stronger rationale for the current study.
- Furthermore, it is unclear whether CAP possesses any distinct advantages or differentiating characteristics compared to other natural polysaccharides in the context of AD.
- The discussion section should provide a more in-depth exploration of the potential mechanisms of CAP in relation to AD pathogenesis, such as its interactions with amyloid-beta and tau pathology.
- While the potential of the PC12 cell model is briefly mentioned, the manuscript lacks discussion on the underlying molecular mechanisms observed in PC12 cells.
- The study explains the therapeutic effects of CAP in AD mainly through intermediary processes like oxidative stress reduction, rather than demonstrating direct effects on AD-specific pathologies. A more focused discussion on CAP’s direct efficacy in AD models is recommended.
- Overall, the structural completeness of the manuscript is lacking. Please revise the reference formatting, include statistical significance indicators in all figures, and organize the figure legends and overall structure more clearly.
Round 2
Reviewer 1 Report
Comments and Suggestions for Authors
All the changes were made to improve the quality of the work.
Reviewer 2 Report
Comments and Suggestions for Authors
The authors have addressed my concerns. I have no further comments.
Reviewer 3 Report
Comments and Suggestions for Authors
The revisions have been successfully reflected. Thank you for your hard work.